

# Enhancing genetic diversity in *Pelargonium*: insights from crossbreeding in the gene pool

Özgül Karagüzel[1], M. Uğur Kahraman[2] and Şevket Alp[3]

[1] Department of Horticulture, Faculty of Agriculture, Recep Tayyip Erdogan University, Rize, Turkey
[2] Department of Vegetables and Ornamental Plants, Batı Akdeniz Agricultural Research Institute, Antalya, Turkey
[3] Department of Landscape Architecture, Faculty of Architecture and Design, Van Yüzüncü Yıl University, Van, Turkey

## ABSTRACT

This study aimed to enrich the *Pelargonium* gene pool through crosses and assess genetic variation among 56 genotypes from five *Pelargonium* species. Seventeen morphological descriptors were used, and NTSYS-pc software was employed to define genetic relationships, and a UPGMA-generated dendrogram reflected these relationships. Moreover, principal component analysis (PCA) was performed to determine which parameter was more effective in explaining variation. Results showed wide variation in genetic similarity rates, with the most similar genotypes being *P. zonale* 'c1' and a hybrid of *P. zonale* 'c1' x *P. zonale* 'c2' (90% similarity). According to the dendrogram results, it was observed that the genotypes were distributed in six clusters. In contrast, the most distant genotypes were *P. zonale* 'c11' and a hybrid of *P. zonale* 'c10' x *P. zonale* 'c11' (0.04% similarity). Hybrids from the female parent *P. x hortorum* 'c1' exhibited unique placement in the dendrogram. In the crossing combinations with this genotype, the individuals obtained in terms of flower type, flower color, flower size, bud size, early flowering, and leaf size characters showed different characteristics from the parents. Surprising outcomes in flower types, colors, and shapes contributed to gene pool enrichment, promising increased breeding variation success. The study holds practical implications for commercial breeding and serves as a valuable guide for future research endeavors.

## INTRODUCTION

In recent years, Pelargonium has become one of the most popular and widely cultivated flowering plants, both in pots and gardens. It is one of the most widely used ornamental plants in home balconies and garden arrangements in recent years. Annually, 500 million *Pelargonium* plants are produced in Europe and 200 million in North America (*International Association of Horticultural Producers (AIPH), 2022*).

*Pelargonium* is the second largest genus with 280 species in 16 sections within the Geraniaceae family and is native to Southeast Africa, Central Asia, Australia, New Zealand, Madagascar, and the South Atlantic Ocean (*Blerot et al., 2018*; *Van de Kerke, 2019*).

Corresponding authors
Özgül Karagüzel,
ozgul.karaguzel@erdogan.edu.tr
M. Uğur Kahraman,
ugurkahramannn@gmail.com

*Pelargonium* and *Geranium* were common genera until the 18th century. Due to their very different flower structures, they were later divided into two genera belonging to the Geraniaceae family. *Pelargonium* species grow in very different ecological conditions; thus, they vary greatly in morphology, anatomy, and cytology. Furthermore, they have different chromosomes. As a result of high diversity, new *Pelargonium* species are constantly being identified (*Plaschil, Abel & Klocke, 2022*). *Pelargonium* species, which were first introduced to Europe in the 17th century, have been among the most widely used indoor and outdoor ornamental plants ever since (*Yu et al., 2016*).

Various *Pelargonium* species have been used to generate hybrids in Europe so far. Among the *Pelargonium* hybrids, the most commercially important species are *P.* x *hortorum*, *P. peltatum*, *P. grandiflorum*, *P. zonale*, and *P. floribunda*. New cultivars continue to be produced yearly due to their popularity (*Plaschil et al., 2012*). *Pelargonium* is one of the top 10 species applying to the CPVO (Community Plant Variety Office) for new cultivar registration in 2021, with 1,084 applications between 1995 and 2021 (*International Union for the Protection of New Varieties of Plants (UPOV), 2022*). Although many new *Pelargonium* cultivars have been developed and commercialized, their genotypic and phenotypic variation is not at the desired level (*Plaschil et al., 2021*). Genetic variation of commercial cultivars is limited (*Plaschil, Abel & Klocke, 2022*).

Increasing genetic diversity is one of the main objectives of breeding programs. Especially in ornamental plants, high genetic diversity is required to gain characteristics such as different flower colors, compactness, long flowering, high number of flowers, and early flowering. If the gene pool is narrow, similar characteristics cannot be deviated from, which means that breeding objectives cannot be achieved.

Genetic diversity or phylogenetic distance can increase spontaneously with climate and environmental influences and natural crosses between species; however, this takes a long time (*Martínez-Cabrera et al., 2012*). The genetic relationship between cultivars significantly impacts the success of the hybridization program, resulting in new cultivars with desired characteristics. On the other hand, the more distant the relationship between the parents, the lower the hybridization success. Plant genetic relationships can be determined by molecular and morphological characterization (*Hartati et al., 2019*). There have been few studies on the morphological characterization of *Pelargonium*. Characterization and evaluation based on morphological characteristics is a quick, easy, and practical guide for parental selection at the hybridization stage. It has been reported that characteristics such as color, shape, flower diameter, number of flowers, flowering time, disease resistance, and growth habit are essential in forming new cultivars by hybridization in *Pelargonium* (*Horn, 1994*; *Gutieva, 2016*).

Further studies are needed to understand the genetic relationships among *Pelargonium* genotypes and to guide breeders in parent selection in future breeding studies. Even current studies show that commercial cultivars are genetically very close to each other. Therefore, increasing genetic diversity is a priority of breeding studies.

New varieties of *Pelargonium* are introduced every year and there is a need in the market. Varieties with different characteristics compared to others are one step ahead in the market. It is necessary to increase variation in this regard. There is a gap in the

literature regarding genetic diversity in *Pelargonium* and methods to enhance this diversity for breeding objectives, and this study aimed to emphasize the extent to which genetic diversity can be increased by crossing.

The present study was conducted to enrich the *Pelargonium* gene pool by crosses and determine the genetic variation among them for the efficiency of the *Pelargonium* breeding program. Moreover, it was aimed to contribute to the existing literature on *Pelargonium* breeding and to shed light on future studies. This study will fill the knowledge gap in the literature by showing the effect of crossing in increasing genetic variation.

## MATERIALS & METHODS

The research was conducted at Bati Akdeniz Agricultural Research Institute, Antalya, Turkey, between 2017 and 2019. The institute is located 17 km from Antalya city center, between 36°56′37″–36°56′39″ North latitude and 30°53′43″–30°53′44″ East longitude. Moreover, its height above sea level is 28 m. The experiment was conducted in a polyethylene greenhouse with a soilless culture system.

A total of 56 *Pelargonium* genotypes, including local cultivars, were used, and the hybrids were generated by hybridization between them. Since hybrid incompatibility was observed in the previously conducted crosses between the parents, the 20 most compatible combinations were selected. The materials used in the study are presented in Table 1.

### Crossing

The crossing was carried out between April 10 and June 7, 2017. Crosses were made among 16 female parents (commercial cultivars) and 20 male parents (18 commercial cultivars and two local cultivars) selected from the gene pool (Table 1). The flowers of female plants were emasculated the day before anthesis and isolated by a paper bag to prevent undesired pollination. Approximately 1–2 days after flower opening, the anthers begin to produce orange-colored pollen grains. A few days after the anthers matured, the five red-purple lobed stigmas began to open and become receptive. Early in the morning pollen was collected from the flower that would serve as the male parent. Collected pollens from male plants were transferred to female plant flower pistils. Crossing was done between 07:00 and 11:00 in the morning. The crosses were made by selecting male flowers with a lot of pollen. After pollination, the pollinated flowers were covered with a plastic bag to prevent uncontrolled pollination. The covering process lasted one week. Seeds were harvested one month after hybridization, and cold stratification was applied. Seeds collected for stratification were put into bags. Seeds were soaked in 2% sodium hypochlorite (NaClO) solution for two minutes and then stored at +4 °C for three months for stratification in perlite. Seeds were sown in September. The germination rate in the crosses varied between 6.3% and 46.7%. The seed growing medium was peat: vermiculite (2/1), germination temperature was 21–23 °C, and humidity was 95–98%. They were then transferred to pots (20 cm) and grown in a greenhouse. The irrigation interval was one minute/hour greenhouse. The lowest temperature in the greenhouse was observed in January with 11.52 °C and the hottest month was July with 32.50 °C. Moreover, the lowest humidity was observed in July with 55.15% and the highest humidity was observed in February with

**Table 1** The materials used in the study.

### GENOTYPES

| No | Name | No | Name |
|---|---|---|---|
| 1 | *P. peltatum* 'c1' | 19 | *P. zonale* 'c13' |
| 2 | *P. peltatum* 'c2' | 20 | *P.* x *hybridum* 'c2' |
| 3 | *P.* x *hortorum* 'c1' | 21 | *P. zonale* 'c14' |
| 4 | *P. interspecific* 'c1' | 22 | *P. zonale* 'c15' |
| 5 | *P.* x *hybridum* 'c1' | 23 | *P.* x *hortorum* 'c2' |
| 6 | *P. zonale* 'c1' | 24 | *P. zonale* 'c16' |
| 7 | *P. zonale* 'c2' | 25 | *P. zonale* 'c17' |
| 8 | *P. zonale* 'c3' | 26 | *P. zonale* 'c18' |
| 9 | *P. zonale* 'c4' | 27 | *P. zonale* 'c19' |
| 10 | *P. zonale* 'c5' | 28 | *P. peltatum* 'c3' |
| 11 | *P. zonale* 'c6' | 29 | L2 |
| 12 | *P. zonale* 'c7' | 30 | *P. zonale* 'c20' |
| 13 | L1 | 31 | *P. zonale* 'c21' |
| 14 | *P. zonale* 'c8' | 32 | *P. zonale* 'c22' |
| 15 | *P. zonale* 'c9' | 33 | *P.* x *hortorum* 'c3' |
| 16 | *P. zonale* 'c10' | 34 | L3 |
| 17 | *P. zonale* 'c11' | 35 | *P. peltatum* 'c4' |
| 18 | *P. zonale* 'c12' | 36 | *P. grandiflorum* 'c1' |

### HYBRIDS

| No | Female | Male |
|---|---|---|
| 37 | *P. peltatum* 'c1' | *P. peltatum* 'c2' |
| 38 | *P.* x *hortorum* 'c1' | *P. interspecific* 'c1' |
| 39 | *P.* x *hortorum* 'c1' | *P.* x *hybridum* 'c1' |
| 40 | *P.* x *hortorum* 'c1' | *P. zonale* 'c16' |
| 41 | *P.* x *hortorum* 'c1' | *P. zonale* 'c17' |
| 42 | *P.* x *hortorum* 'c1' | *P. zonale* 'c20' |
| 43 | *P. zonale* 'c1' | *P. zonale* 'c2' |
| 44 | *P. zonale* 'c3' | *P. zonale* 'c4' |
| 45 | *P. zonale* 'c5' | *P. zonale* 'c6' |
| 46 | *P. zonale* 'c7' | L1 |
| 47 | *P. zonale* 'c8' | *P. zonale* 'c9' |
| 48 | *P. zonale* 'c10' | *P. zonale* 'c11' |
| 49 | *P. zonale* 'c12' | *P. zonale* 'c13' |
| 50 | *P.* x *hybridum* 'c2' | *P. zonale* 'c14' |
| 51 | *P. zonale* 'c15' | *P.* x *hortorum* 'c2' |
| 52 | *P. zonale* 'c18' | *P. zonale* 'c19' |
| 53 | *P. zonale* 'c3' | L2 |
| 54 | *P. zonale* 'c21' | *P. zonale* 'c22' |
| 55 | *P.* x *hortorum* 'c3' | L3 |
| 56 | *P. peltatum* 'c4' | *P. grandiflorum* 'c1' |

**Table 2   Descriptors used for characterization and evaluation of *Pelargonium* genotypes.**

| Characteristics | Description |
|---|---|
| Flower number (per plant) | Score range (1 = Many 3 = Medium, 5 = Few) |
| Flower stem length (cm) | Score range (1 = Long, 3 = Medium, 5 = Short) |
| Flower type (number) | Score range (1 = Double, 3 = Semi-double, 5 = Single) |
| Flower width (cm) | Score range (1 = Broad, 3 = Medium, 5 = Narrow) |
| Leaf size (cm) | Score range (1 = Broad, 3 = Medium, 5 = Narrow) |
| Length of internodes (cm) | Score range (1 = Long, 3 = Intermediate, 5 = Short) |
| Plant height (cm) | Score range (1 = Long, 3 = Medium, 5 = Short) |
| Bud size (unopened) | Score range (1 = Broad, 3 = Medium, 5 = Narrow) |
| Earliness | Score range (1 = Very early, 3 = Early, 5 = Intermediate, 7 = Late) |
| Flower color | Score range (1 = Light 3 = Middle 5 = Dark) |
| Growth habit | Score range (1 = Upright, 3 = Semi trailing, 5 = Trailing) |
| Growth vigor | Score range (1 = Compact, 3 = Semi compact, 5 = Strong) |
| Leaf color | Score range (1 = Light green, 3 = Green, 5 = Dark green) |
| Leaf hairiness | Score range (1 = Dense, 3 = Intermediate, 5 = Tenuous, 7 = Absent) |
| Stem color | Score range (1 = Green, 3 = Whitish green, 5 = Brown red) |
| Stem hairiness | Score range (1 = Dense, 3 = Intermediate, 5 = Tenuous, 7 = Absent) |
| Stem thickness | Score range (1 = Thick, 3 = Intermediate, 5 = Thin) |

65.28%. Furthermore, when the average day length data was evaluated, it was observed that there was an annual day length of 12.16 h, the shortest day was observed in January with 10.22 h and the longest day was observed in June with 14.60 h.

## Morphological observations

Observations of 56 genotypes were made on a total of 17 characters, seven quantitative and ten qualitative, using some plant trait criteria (Table 2) of the International Union for the Protection of New Varieties of Plants (*International Union for the Protection of New Varieties of Plants (UPOV), 2009*) and breeders. Morphological observations of quantitative and qualitative characters were taken when the plant reached maturity for sale in the market. Quantitative characteristics were flower number, flower stem length, flower type, flower width, leaf size, length of internodes, and plant height. Qualitative characteristics were bud size, earliness, flower color, growth habit, growth vigor, leaf color, leaf hairiness, stem color, stem hairiness, and stem thickness. Methods for measuring quantitative characteristics of the plant, leaf, and flower are given below.

1. Flower number: It was calculated as the number of open flowers.
2. Flower stem length (cm): It was measured from the lower petal to the base from the lower.
3. Flower type (number): It was calculated that a single flower has five petals, a double flower has more than ten petals, and a semi-double flower has 5-10 petals.
4. Flower width (cm): It was measured as the length of the largest flower.

5. Leaf size (cm): It was recorded as the longest extension from leaf apex to base (*i.e.,* the connection point of the leaf blade and petiole).
6. Length of the internodes (cm): It was calculated by measuring the spacing of the two nodes on the most vigorous stem.
7. Plant height (cm): Plants were measured from ground level (the base of the plant) to the tip of the highest point.

## Statistical analysis

Morphological observations were taken in three replicates, with five plants in each replicate. The plants were planted at the same time, and their development was parallel. Genetic similarity was analyzed by the UPGMA (unweighted pair-group method, arithmetic average) clustering procedure using the NTSYS (Numerical Taxonomy Multivariate Analysis System) software (Version: 2.2) (*Rohlf, 2000*) using morphological data. Excel software was used to calculate the means and standard deviation to explain the variation in the population. After all data were standardized, PCA was performed using JMP statistical software to determine which parameter was more effective in explaining variation.

# RESULTS

A total of 17 morphological descriptors were used to determine the phylogenetic relationships. High morphological diversity was observed between *Pelargonium* genotypes characterized by quantitative and qualitative properties. The Eigenvalue was 98.35. The higher the Eigenvalue, the higher the variance. The approximate Mantel t-statistic test was $t = 12.4355$ and $p = 1.0000$. The matrix correlation (r) was 0.50. The similarity rates according to the coefficient similarity of genotypes ranged between 0.04 and 0.90. According to the dendrogram results, it was observed that the genotypes were distributed in six clusters.

## The dendrogram of cluster analysis of *Pelargonium* genotypes

Two major groups were revealed with a dendrogram generated by the UPGMA method using morphological data (Fig. 1). In the first group, one genotype was in a separate group with a 0.04% similarity rate. This genotype was obtained from the crosses among commercial cultivars *P. zonale* 'c10' x *P. zonale* 'c11'. Furthermore, the second group was divided into two subgroups. In the second group (among 55 genotypes), 14 genotypes were included in one group, and 41 were included in the other group. The highest genetic similarity was observed between *P. zonale* 'c1' x *P. zonale* 'c2', *P. zonale* 'c3' x *P. zonale* 'c4', *P.* x *hortorum* 'c1' x *P. zonale* 'c17', *P. zonale* 'c18' x *P. zonale* 'c19', and *P. zonale* 'c21' x *P. zonale* 'c22' hybrids with their male parents and between the female parent and *P. zonale* 'c5' x *P. zonale* 'c6' hybrids. These hybrids carried the characteristics of their parents. The genotype with the most distant genetic diversity with their parents was observed in one of the *P. zonale* 'c10' x *P. zonale* 'c11' hybrids. Moreover, hybrids obtained from crosses of female parent *P.* x *hortorum* 'c1' and five male parents were separated into different groups. These genotypes enriched the gene pool due to the increased genetic variation. The genetic similarity rates showed a wide variation among the genotypes. The most similar genotypes

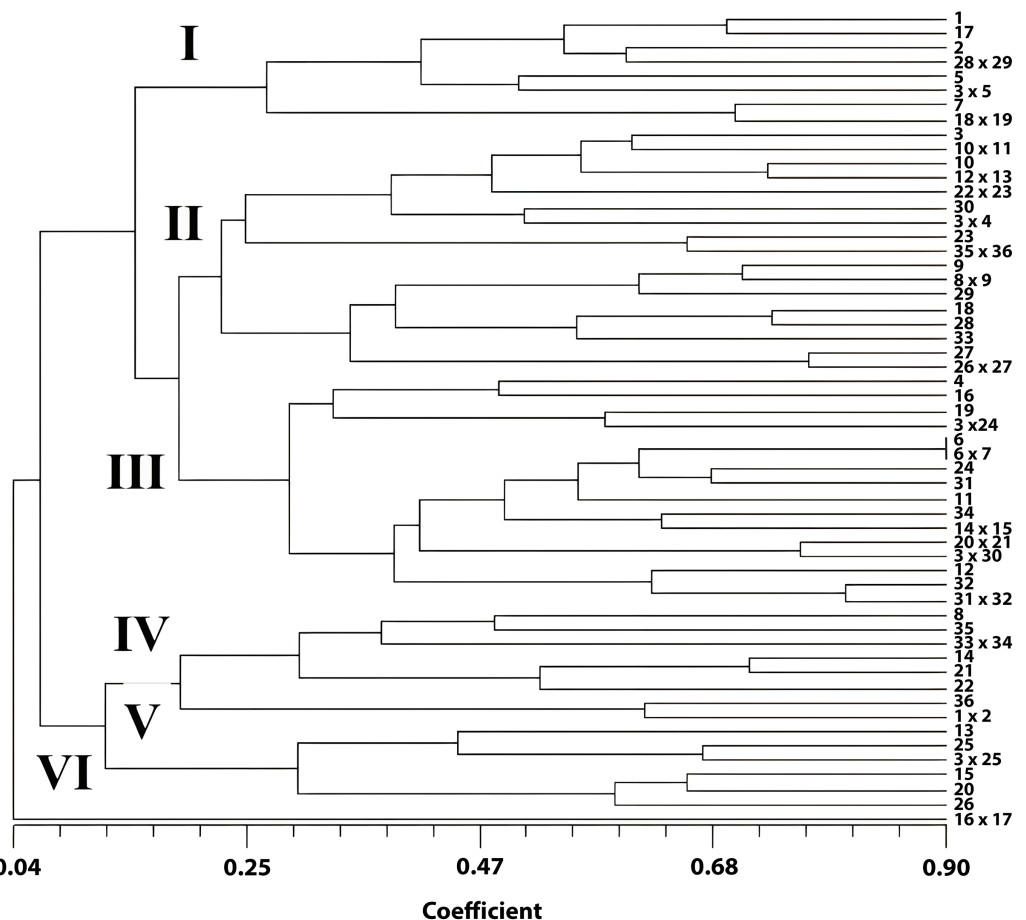

**Figure 1** UPGMA dendrogram showing phylogenetic relationships of *Pelargonium* genotypes.

were *P. zonale* 'c1' and a hybrid of *P. zonale* 'c1' x *P. zonale* 'c2' with a 90% similarity. However, the most distant genotypes were *P. zonale* 'c11' and a hybrid of *P. zonale* 'c10' x *P. zonale* 'c11' with 0.04% similarity. The hybrids, generated from the female parent *P. x hortorum* 'c1', were separated differently in the dendrogram.

## Principal component analysis

The distributions of both genotypes and parameters were analyzed in the coordinate plane shown in Fig. 2. According to Fig. 2, the origin in the coordinate plane describes the least variation. Moreover, the close positioning of parameters or genotypes indicates a high degree of similarity between them. Some of the parameters used in this study were close to the origin while others were far from it. According to PCA, length of internodes (eigenvalue of 3.04), flower type (eigenvalue of 2.62), flower stem length (eigenvalue of 1.87), flower width (eigenvalue of 1.37), and leaf size (eigenvalue of 1.18) explained 59.35% of the total variation. PCA results showed that the variation was wide and supported the dendogram. According to the Fig. 2, earliness and stem color parameters showed similar effects. Furthermore, the flower color and growth habit parameters and the leaf hairiness

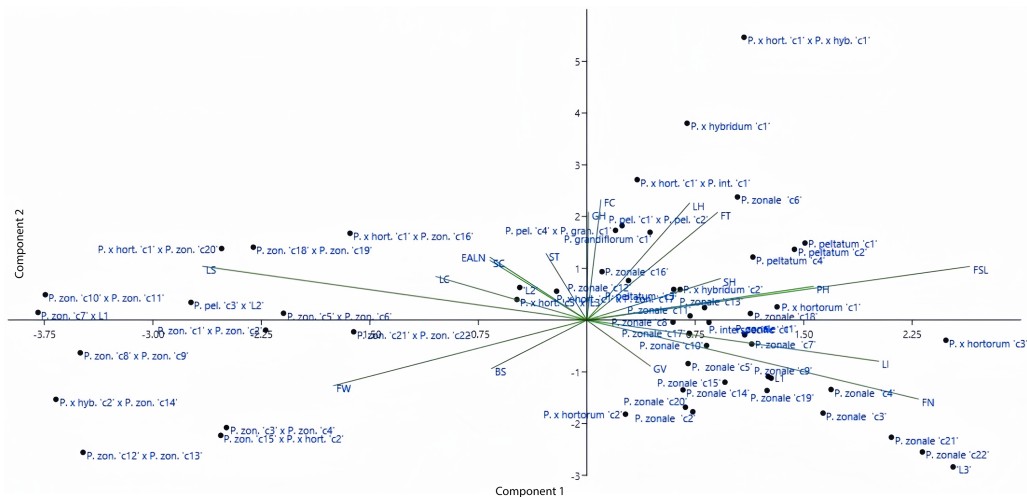

**Figure 2   Distribution of the investigated parameters and employed genotypes in principal coordinates.** LI, Length of Internodes; PH, Plant Height; FT, Flower Type; FSL, Flower Stem Length; LS, Leaf Size; FW, Flower Width; FN, Flower Number; LC, Leaf Color; LH, Leaf Hairiness; FC, Flower Color; BS, Bud Size; EALN, Earliness; ST, Stem Thickness; SH, Stem Hairiness; SC, Stem Color; GH, Growth Habit; GV, Growth Vigor.

and flower type parameters were located close to each other. Looking at the distribution of genorypes, it can be seen that the variation was wide. Although the genotypes belonging to the same species were in close distance to each other, the hybrid individuals were located farther away from the parents. This shows the effect of crossing in increasing variation.

## Variation of characteristics

The similarities between the genotypes used as parents and the hybrid individuals obtained in terms of the morphological characters examined are given in detail in Table S1. As can be seen in Table S1, according to the remarkable characteristics obtained from *P. peltatum* 'c1' x *P. peltatum* 'c2' hybrid to create variation, although the leaf color of the maternal parent was medium and the paternal parent was dark, the hybrid individual was in the light group. While the leaves of the maternal parent were absent and the paternal parent was tenuous, the hybrid showed a medium level of hairiness. Moreover, in terms of plant height, while the parents were medium-sized, the hybrid was short. In another hybridization made for the same purpose, in the combination of *P.* x *hortorum* 'c1' x *P. interspecific* 'c1', the petal shapes of the parent individuals were semi-double, but the hybrids showed double characteristics. In the same combination, flower color was medium in the parents, but the hybrids showed dark coloration. Likewise, the parents' flower size was 3–4 cm (intermediate), while the hybrid obtained reached 5–6 cm (large). Furthermore, leaf size was different from the parents. In another combination, *P.* x *hortorum* 'c1' x *P.* x *hybridum* 'c1', although the parents had semi-double flowers, the hybrid plant showed double flower characteristics. Bud size was determined to be medium in the parents but large in the hybrid plant. Early flowering was observed in the hybrids in the same combination, while the parents showed intermediate earliness. According to the results obtained from *P. zonale*

'c3' x *P. zonale* 'c4' cross combination, flower size was 3–4 cm (intermediate) in the parents, while 5–6 cm (large) was obtained from the hybrid. Moreover, in *P. zonale* 'c5' x *P. zonale* 'c6', different results were obtained in flower size from the parent traits. The flowers of the hybrid plants were 1–2 cm in size, while the maternal and paternal parents had 3–4 cm and 5–6 cm flowers, respectively. In the *P. zonale* 'c7' x L1 combination, although the maternal parent was double-flowered and the paternal parent was single-flowered, the hybrids showed semi-double characteristics. Flower size also showed variation in the same combination. The flowers of the hybrid plants were small (1–2 cm), while the maternal and paternal parents had intermediate (3–4 cm) and large (5–6 cm) flowers, respectively. The hybrid *P. zonale* 'c12' x *P. zonale* 'c13' showed double flowers while the parents were semi-doubled. Furthermore, the parents had light-colored leaves in the combination of *P. x hybridum* 'c2' x *P. zonale* 'c14', whereas the hybrid showed dark-colored leaves. In the same combination, the parents were single-flowered, while the hybrid had semi-double flowers. In the combination of *P. x hortorum* 'c1' x *P. zonale* 'c16', the maternal parent had dark, the paternal parent had medium, and the hybrid had light-colored flowers. In the hybrid of *P. x hortorum* 'c1' x *P. zonale* 'c17', the buds of the parents were medium, while the hybrid obtained was small. Furthermore, the bud size of the parents of *P. zonale* 'c18' x *P. zonale* 'c19' was medium, while the hybrid was large. While *P. peltatum* 'c3' x L2 hybrid showed a double-flowered character, the maternal parent showed a single character, and the paternal parent showed a semi-double character. *P. zonale* 'c21' x *P. zonale* 'c22' hybrids showed variation in plant height. While the parents were medium-sized, the hybrid plants were short. According to the results of *P. peltatum* 'c4' x *P. grandiflorum* 'c1' combination, the leaf color of the parents was light, while the hybrid was dark. The flower structure and petal color characteristics of parent plants and hybrids are shown in Fig. 3. As expected, the dark color of the petals was detected as dominant in some hybrids. For instance, the hybridization of an orange-colored female (*P. zonale* 'c5') and a white-colored male (*P. zonale* 'c6') resulted in an orange-colored hybrid. Likewise, an imperial red hybrid was obtained by crossing a red-colored male genotype (*P. x hybridum* 'c1') with a salmon female genotype (*P. x hortorum* 'c1'). However, surprising results were also obtained from some crosses. For instance, *P. zonale* 'c18' x *P. zonale* 'c19' and *P. zonale* 'c21' x *P. zonale* 'c22' hybrid flower colors were lighter than their parents' dark flowers.

## DISCUSSION

The hybrid individuals' morphological characteristics were similar to those of both parents according to the combination and trait. However, there were also hybrid individuals showing different characteristics from the parents. Genetic variation studies in ornamental plants have been carried out in various species in previous studies. In their two studies on cyclamen, *Curuk et al. (2015)* and *Curuk et al. (2016)* performed morphological characterization among *Cyclamen* species naturally distributed in Turkey and evaluated the population in terms of genetic variation. In the study, high variation was determined in terms of many characteristics and the importance of hybrid studies to be carried out in the coming years was emphasized. *Singh et al. (2018)* observed high variation in some

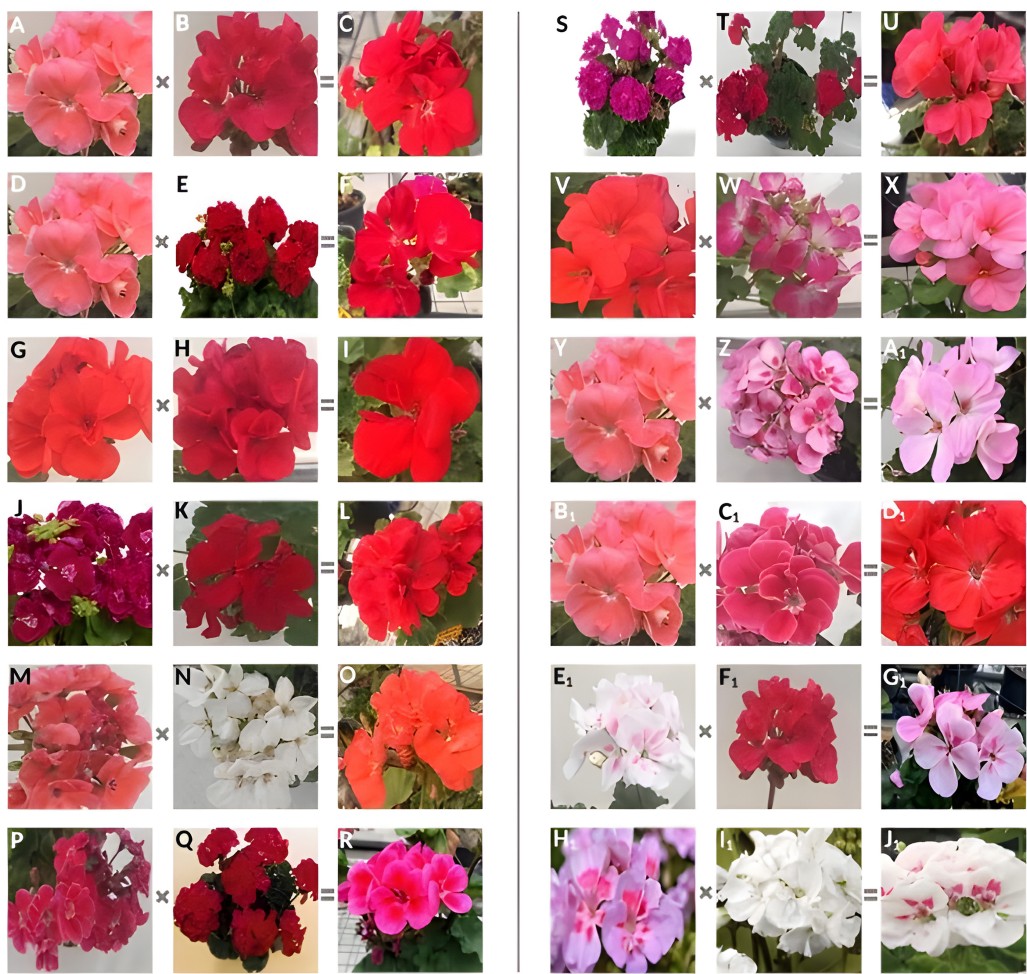

**Figure 3** **Appearance of flower color differences of some hybrids and their parents.** (A) *P*. x *hortorum* 'Survivor Salmon Sensation'; (B) *P. interspecific* 'Sarita Punch'; (C) hybrid; (D) *P*. x *hortorum* 'Survivor Salmon Sensation'; (E) *P*. x *hybridum* 'Calliope Dark Red'; (F) hybrid; (G) *P. zonale* 'Savannah Oh So Orange'; (H) *P. zonale* 'Survivor Cherry Red'; (I) hybrid; (J) *P. zonale* 'Pac Flower Fairy White Splash'; (K) *P. zonale* 'Senna Red'; (L) hybrid; (M) *P. zonale* 'Toscana Sil Gatwig'; (N) *P. zonale* 'Toscana Sil Wenke'; (O) hybrid; (P) *P. zonale* 'Toscana Tammo'; (Q) *P. zonale* 'Toscana Sil Onno'; (R) hybrid; (S) *P. zonale* 'Toscana Sil Merle'; (T) *P. zonale* 'Pac Samelia'; (U) hybrid; (V) *P. zonale* 'Pac Stefanie'; (W) *P*. x *hortorum* 'Ringo 2000 Rose'; (X) hybrid; (Y) *P*. x *hortorum* 'Survivor Salmon Sensation'; (Z) *P. zonale* 'Savannah Pink Mega Splash'; (A1) hybrid; (B1) *P*. x *hortorum* 'Survivor Salmon Sensation'; (C1) *P. zonale* 'Claudio DCM'; (D1) hybrid; (E1) *P. zonale* 'Svannah White Splash'; (F1) *P. zonale* 'Savannah Punch'; (G1) hybrid; (H1) *P. zonale* 'Grandeur Classic Light Pink Splash'; (I1) *P. zonale* 'Grandeur Power White'; (J1) hybrid. Photo credit: Özgül KARAGÜZEL.

characteristics in their study on 50 *Gladiolus* cultivars. At the end of the study, it was reported that these varieties with genetic variation among each other will be a good genetic resource in future breeding studies. *Chen et al. (2019)* conducted a study with 14 natural *Lilium* species and eight hybrids and obtained a wide variation between natural species and hybrids. In the study, it was emphasized that future hybridization studies may be possible. *Rastogi et al. (2019)* investigated genetic variability both molecularly and morphologically

in 48 *Bougainvillea* cultivars obtained by different methods such as hybridization. In the study, a wide range of genetic distance between *Bougainvillea* varieties was observed. This genetic distance revealed that there was a wide variety and it can be further expanded with future hybridization studies. *Azimi & Alavijeh (2020)* observed high variation in 16 amaryllis (*Hippeastrum hybridum*) genotypes, especially in some characteristics such as flowering period and petal length. As a result of the study, it was reported that genetic factors caused wide variation among genotypes. *Rêgo et al. (2020)* studied genetic differences in 12 specimens belonging to the genus Sansevieria. They determined genetic differences in leaf color, shape and size. As a result of the study, it was reported that the UPGMA method was an effective method for determining variation. *Taghipour et al. (2021)* observed variation in 21 characters in 30 dendranthema (*Dendranthema grandiflorum*) cultivars. In the study, 11 genotypes were clustered in four different groups (cluster), while the other 19 genotypes were clustered in four other groups. As a result of the study, it was predicted that successful hybrid studies could be carried out in the future among individuals with high variation.

In some studies about *Pelagonium* where genetic similarity was analyzed, it was determined that genetic similarity rates differed between genotypes. While the genetic variation in *Pelargonium crispum* and hybrid cultivars was reported as 30% (*Plaschil et al., 2016*), the genetic similarity rate ranked between 0.04 and 0.90 in this study. This result revealed the value of the created gene pool by hybridization.

It was stated that characteristics such as color, shape, flower diameter, number of flowers, flowering time, disease resistance, and growth habit are important in forming new cultivars with hybridization studies in *Pelargonium* (*Gutieva, 2016*). Additionally, some researchers investigated color inheritance in the flowers of zonal Pelargonium species. According to the researchers, the inheritance of the red flower color in *Pelargonium* species is controlled by the interaction of two alleles genes, R1/r1 and R2/r2. In the absence of one or both dominant genes, the flower has a non-red color. When a plant has a genotype with R2 and without R1, the flower color becomes salmon, and when R1 is present and there is no R2, the flower color is rose pink. The flower color is soft pink if a plant has a genotype with R1 and R2 missing. Moreover, the orange color is due to the homozygous recessive alleles, and when orange-flowered plants were hybridized with white-flowered plants, peach-colored flowers were obtained in the F1 generation (*Nasser & Tilney-Bassett, 1992*; *Tilney-Bassett et al., 1995*). Although similar results were also obtained in the present study, unexpected hybridization, like obtaining white color from orange and white crosses, also occurred. In addition, it was observed that the paternal contribution to the color of the flowers of F1 hybrids was higher compared to the maternal in this study. In a previous study *Plaschil et al. (2021)*, interspecies hybridization was conducted to transfer the bright red color of *P. fulgidum*. However, it was stated that the obtained diploid hybrids had undesirable morphological characteristics. Moreover, biotechnological approaches such as embryo rescue are needed to obtain $F_1$ hybrid individuals from *P. fulgidum*. The findings obtained from the present study show that it is possible to obtain desired red color shades with species such as *P.* x *hortorum* x *P. zonale* and *P.* x *hortorum* x *P.* x *hybridum* by choosing the appropriate parent for the conventional hybridization.

*Pelargonium* cultivars have a significant portion in the ornamental plants market in Europe and North America. Breeders are trying to create new characteristics through interspecific hybridization in the increasingly narrowing gene pool to maintain its high market position. For this purpose, they are searching for new hybrids with more desirable characteristics through embryo rescue techniques in species with low success in crossing. However, promising results that can be commercialized have not yet been achieved (*Kamlah et al., 2019*). The finding of this study showed that the most important *Pelargonium* species, which are still used for the development of commercial cultivars, are still capable of developing new favorable characteristics.

Phenotyping is the simplest tool for selection used in breeding programs and allows the description to define the phenotypic value of progenies of selected parents. The phenotyping approach has been used in the selection of crops for more than a century and has been gradually developing. In ornamental breeding, phenotyping is still limited to easily scored characteristics, such as petal and leaf color or growth type, based on the breeder's experience. In *Pelargonium*, which has a rich genetic diversity, using analytical methods to determine the genotypic variance of the population in phenotyping studies and to make a good selection is quite beneficial for developing promising cultivars (*Molenaar et al., 2017*). If breeders take into account the genetic distance between the parents in their experiments, it is possible for them to obtain more successful results in crosses (*Akbarzadeh et al., 2023*). In this study, hybrids obtained from interspecific crosses between commercial cultivars and local genotypes of different *Pelargonium* species were phenotyped for some important characteristics. Moreover, phylogenetic information obtained from the morphological data provided useful information for breeders in selection.

There are very few studies on the heritability of floral traits in *Pelargonium*, and our study has contributed to the literature. In addition to the $F_1$ population, the individuals obtained in the study can be selfed in future studies. Thus, the $F_2$ population can be obtained, and studies can be carried out to reveal the heredity more clearly. In addition, commercial companies can increase the number of *Pelargonium* plants in their gene pools and apply interspecific hybridization methods. In addition, methods such as embryo rescue can be used to increase the success of interspecific hybridization. In particular, the most commercially important morphological traits can be examined in depth, and heritability studies can be conducted for each morphological trait separately.

## CONCLUSIONS

The knowledge of genetic distance and variability is essential for plant breeding and interspecific hybridization. The high variation among the quantitative characters measured in the genotypes studied showed that this situation can give a good idea for achieving the desired characteristics for developing promising *Pelargonium* cultivars. The main conclusion that can be drawn is that hybridization in constructed gene pools for breeding studies is a highly effective way to increase genetic diversity. This is a crucial point in future attempts to overcome the problems caused by the narrowness of the genetic base. As a result of the study, it was determined that there was genetic variation among *Pelargonium*

genotypes and this variation increased with crossing. There are limited studies in the literature on increasing *Pelargonium* genetic variation by crossing. In this sense, it is thought that the present study will contribute to the literature. In future studies, a different hybridization program can be designed in *Pelargonium* and different species can be added to diversify the variation. The trend in ornamental plants and *Pelargonium* is changing every day. Different colors, types, sizes and shapes of flowers and leaves, and different types and heights of plants are being introduced to the sector. This study demonstrates that with a properly planned crossing program, *Pelargonium* plants with different characteristics that will bring innovation to the sector can be introduced.

The study demonstrated the importance of crossing in *Pelargonium* to increase genetic variation. However, the fact that the study was carried out with classical methods was limiting for heritability. In future studies, molecular techniques can be used to examine certain morphological traits in more detail. Genetic mapping, gene expression, and the development of molecular markers for specific traits are suggested for future studies. Furthermore, it is very important for commercial companies to keep their gene pools as large as possible, increase the number of hybridizations to keep the market alive in *Pelargonium*, and develop new cultivars continuously.

## ACKNOWLEDGEMENTS

The authors thank the Bati Akdeniz Agricultural Research Institute and the General Directorate of Agricultural Research and Policies. Moreover, we would like to thank Assoc. Prof. Dr. Hatice Filiz BOYACI and Dr. Esra CEBECI.

### Funding

This study was supported by the General Directorate of Agricultural Research and Policies, Re-public of Turkey Ministry of Agriculture and Forestry, under Project No: TA-GEM/BBDA/16/A09/P08/01. The funders had no role in study design, data collection and analysis, decision to publish, or preparation of the manuscript.

### Grant Disclosures

The following grant information was disclosed by the authors:
General Directorate of Agricultural Research and Policies, Re-public of Turkey Ministry of Agriculture and Forestry: TA-GEM/BBDA/16/A09/P08/01.

### Competing Interests

The authors declare there are no competing interests.

### Author Contributions

- Özgül Karagüzel conceived and designed the experiments, performed the experiments, analyzed the data, authored or reviewed drafts of the article, and approved the final draft.

- M. Uğur Kahraman conceived and designed the experiments, performed the experiments, prepared figures and/or tables, authored or reviewed drafts of the article, and approved the final draft.
- Şevket Alp performed the experiments, authored or reviewed drafts of the article, and approved the final draft.

### Data Availability

The raw data are available in the Supplementary File.

### Supplemental Information

Supplemental information for this article can be found online at http://dx.doi.org/10.7717/peerj.17993#supplemental-information.

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
