# Peer review of "Enhancing genetic diversity in Pelargonium: insights from crossbreeding in the gene pool"

_PeerJ, doi:10.7717/peerj.17993_

## Round 0.1 · original submission · Minor Revisions

Your study contains important information regarding the breeding of an important globally widespread ornamental plant, and your presentation is clear and concise. However, it needs to be improved. Complete minor revisions by addressing the following deficiencies in addition to the reviewers' comments. Even though one of the reviewers marked it as a major revision, when I read the comments I saw minor revisions. Since I believe that you will complete the changes I want in a short time, my decision is for minor revisions.

-In the Materials and Method section, you wrote that you chose 16 female and 20 male parents, but there are 20 hybrids made between them. When I read this I expect a 16 x 20 crossover. But you did not crossbreed them all with each other. How did you decide to make these crossbreeds? You must explain.
-It will be more understandable if you show male and female parents in separate columns and hybrids in a separate column in Table 1. It's hard to keep track like that.
-Table 3 is quite long and does not look good in shape. Supp it. I recommend you present it as a file. Or you can prepare separate tables for male parent, female parent and hybrids.
-I recommend that you add a table showing the genetic distance values ​​of the genotypes or writing down the members of each group. It is difficult to follow from the dendrogram. You should also improve its appearance. It would be better if you create a circular dendogram.

·

Basic reporting

English usage: The article is written in generally understandable English. Professional standards of courtesy and expression are adhered to.

Literature references, field background/context provided: The introduction section of the article needs to be restructured for better flow. There are sentences in the introduction section that repeat unnecessarily. The importance of the Pelargonium plant is elaborated for too long. It should be expressed more concisely and clearly. Additionally, the purpose of the study and why this research is needed should be emphasized more broadly in the introduction section. Similar literature studies should be discussed, and the differences should be explained.

Raw data sharing: Raw data has been shared. It is understandable.

Article structure, figures: The structure of the article conforms to a standard format with appropriate sections. The figures are relevant to the content of the article, but improving the quality of both figures would be better. Attention should be paid to the Latin spelling of species names in Figure 1.

Results and Hypothesis-Based Evaluation: The submitted study contains results relevant to the hypothesis, but there are aspects that need improvement, as mentioned in the subsequent sections. The article has a coherent structure.

Experimental design

Relevance to the Journal's Scope: The article represents an original primary research within the scope of the journal.

Research Question: The research question is well-defined, relevant, and meaningful. However, it could be improved. Additionally, it should be clearer how the study fills the identified knowledge gap.

Technical and Ethical Standards: The research has been conducted meticulously and adheres to standard technical and ethical practices.

Methodology: It is important to reference the literature regarding the methods used. The methodology section should be enhanced for better comprehensibility and reproducibility. For example, the following questions could be addressed:
1- What is the appropriate flower form for crossbreeding?
2- At what time of the day was crossbreeding conducted?
3- For how many days were the flowers kept under the paper bag after pollination?
4- What are the qualitative characteristics?
5- How many crossbreedings were conducted in total? How many hybrids were obtained?
6- What was the germination rate? Were all hybrids used? If not, how were they selected?

Validity of the findings

This article extensively evaluates various morphological features among Pelargonium genotypes. It demonstrates various analyses conducted to determine genetic similarities and differences among Pelargonium genotypes, as well as examines the characteristics of hybrids resulting from crossbreeding. In terms of results evaluation, it is evident that appropriate statistical methods were used to identify genetic similarities and differences among genotypes. Additionally, changes in the characteristics of hybrids resulting from crossbreeding are thoroughly examined and compared with their parents. The presented results appear noteworthy and contribute to understanding the genetic diversity and morphological features of Pelargonium species. The discussion section seems quite informative. The researchers indicate that it is possible to develop new traits through interspecific hybridization of Pelargonium species. They highlight that these hybridizations create diversity in genetic variation and phenotypic traits, enabling the development of important characteristics. Moreover, they make recommendations on how interspecific hybridization methods and phenotyping techniques can be adapted for future research and commercial applications. However, I suggest that the discussion section should reference other research findings more extensively and evaluate their findings within a broader literature context. Particularly, discussing how the obtained findings align or differ with the results of similar studies on Pelargonium species, by referencing other studies on Pelargonium species, would be beneficial.

Additional comments

No comments.

Reviewer 2 ·

Basic reporting

I have reviewed the provided manuscript, "Enhancing genetic diversity in Pelargonium: insights from crossbreeding in the gene pool," based on the detailed criteria you outlined. Below is a comprehensive comments:
The manuscript is written in professional English but contains minor grammatical errors and awkward phrasing that could be improved for better readability and clarity.
 Line 19: "The study aimed to enrich..." can be rephrased to "This study aimed to enrich..."
 Line 79: "The genetic relationship between cultivars is quite effective..." should be "The genetic relationship between cultivars significantly impacts..."
 Line 25: "Using 17 morphological descriptors, NTSYS-pc software was employed to define genetic relationships..." can be rephrased to "Seventeen morphological descriptors were used, and NTSYS-pc software was employed to define genetic relationships..."
 Line 93: "There is a gap in the literature on genetic diversity in Pelargonium and how this diversity can be increased to achieve breeding objectives," could be rephrased to "There is a gap in the literature regarding genetic diversity in Pelargonium and methods to enhance this diversity for breeding objectives."
Some sentences could be simplified or restructured for clarity:
 Line 41: "In recent years, Pelargonium has become one of the most popular and widely cultivated species of all flowering potted plants and garden plants." This could be rephrased to "In recent years, Pelargonium has become one of the most popular and widely cultivated flowering plants, both in pots and gardens."
 Line 45: "Although the plant is associated with the Mediterranean region, it is very popular all over the world." This could be rephrased to "While traditionally associated with the Mediterranean region, Pelargonium is now popular worldwide."
The introduction provides a comprehensive overview of the importance of Pelargonium in horticulture and its commercial value.
The study's significance is well-articulated, emphasizing the need for genetic diversity in breeding programs.
The literature review is thorough but could benefit from the inclusion of more recent studies to provide an updated contex. The research gap is clearly identified, and the study aims to address this gap by enhancing genetic diversity through crossbreeding.

Literature references, sufficient field background/context provided:
The introduction provides a thorough background on Pelargonium, its commercial importance, and the necessity for genetic diversity in breeding programs.
While the manuscript cites relevant literature, some references are outdated (e.g., Horn, 1994). Including more recent studies would provide updated context and strengthen the manuscript:
Studies from the last five years on genetic diversity in ornamental plants could enhance the literature review.
The literature review covers a broad range of studies but could be improved by including more recent developments in crossbreeding and genetic diversity research. This would provide a more balanced view of the current state of the field.
The manuscript follows the standard format with clearly defined sections: Introduction, Materials & Methods, Results, Discussion, and Conclusions.
Figures and Tables:
 Figure 1: The dendrogram effectively illustrates the genetic relationships among the Pelargonium genotypes.
 Table 1: Clearly lists the genotypes and hybrids used, providing a comprehensive overview of the materials.
 Table 2: The descriptors for morphological characterization are well-defined and standard.
 Table 3: Provides a detailed distribution of morphological characteristics.
The raw data is shared in accordance with the journal's policy. It is crucial to ensure that the data is easily accessible and clearly documented for reproducibility.
Self-contained with relevant results to questions:
The manuscript is self-contained and presents a coherent study that directly addresses the research questions posed in the introduction. The results are relevant and linked to the hypotheses and objectives stated at the beginning.

Experimental design

Experimental Design
The research is original and fits well within the scope of the journal, focusing on enhancing genetic diversity through crossbreeding, which is a significant area of interest in plant breeding.
Research question well defined, relevant & meaningful. It is stated how research fills an identified knowledge gap:
Clarity and Justification: The research question is clearly defined. The study aims to enhance genetic diversity in Pelargonium through crossbreeding, filling a notable gap in current knowledge.
The manuscript effectively conveys the significance of this research in the context of commercial breeding and genetic diversity.
Rigorous investigation performed to a high technical & ethical standard:
Methodological Rigor: The investigation employs standard crossbreeding techniques and robust genetic analysis.
Ethical Standards: Ethical standards are maintained in the handling of plant material. Specific ethical guidelines or approvals required for this study should be mentioned if applicable.
Potential Bias: The manuscript should discuss any potential sources of bias in the experimental design and how they were mitigated.
Materials and Methods
The materials used in the study, including the 56 Pelargonium genotypes, are clearly describe. The crossbreeding techniques are detailed, including emasculation, pollination, and seed stratification processes.
o Environmental Conditions: Additional details on the greenhouse environment, such as light levels, humidity, and temperature, would enhance the reproducibility of the study.
o Morphological Observations: The methods for morphological observations are well-defined, using standard descriptors from the International Union for the Protection of New Varieties of Plants (UPOV).
o Statistical Analysis: The statistical methods, including the use of NTSYS-pc software and UPGMA clustering, are appropriate for the study's objectives.
Minor clarifications could be added regarding the specific conditions of the greenhouse environment (e.g., light levels, humidity, temperature) and any variations that might have occurred.
The manuscript should mention any specific challenges or deviations from the planned methodology and how they were addressed.

Validity of the findings

Impact and novelty not assessed. Meaningful replication encouraged where rationale & benefit to literature is clearly stated:
The study encourages replication by providing a clear rationale and discussing the benefits of increasing genetic diversity in Pelargonium.
Benefits to Literature: The potential benefits to the literature and the broader field are well-articulated.
All underlying data have been provided; they are robust, statistically sound, & controlled:
The data provided are robust and statistically sound, with appropriate controls. The statistical methods used, such as the Mantel test and UPGMA clustering, are suitable for the study's objectives. The manuscript should consider whether additional statistical methods could further validate the findings. For example, multivariate analysis or bootstrap analysis to confirm the stability of the clusters.
Conclusions are well stated, linked to original research question & limited to supporting results:
The conclusions are well-stated and linked to the original research questions.
The conclusions adequately address the limitations of the study. Future research directions are suggested based on the findings.
The manuscript should discuss the broader implications of the findings, particularly for commercial breeding programs and genetic diversity enhancement.

Additional comments

General Comments
Strengths:
• Comprehensive Background: The introduction provides a thorough background and sets the stage for the research.
• Detailed Methodology: The methods section is detailed and enables reproducibility.
• Clear Presentation: The results are presented clearly, supported by relevant figures and tables.
Weaknesses:
• Language and Grammar: Minor grammatical errors and awkward phrasings should be corrected to improve readability.
• Updated References: Some references are outdated. Including more recent studies would provide updated context.
• Environmental Details: Additional details on the environmental conditions in the greenhouse could enhance reproducibility.
Results
• Presentation of Data: The results are presented clearly and logically, with relevant figures and tables supporting the findings.
o Genetic Relationships: The dendrogram effectively illustrates the genetic relationships among the Pelargonium genotypes.
o Morphological Diversity: High morphological diversity is observed among the genotypes, which is well-documented in Table 3.
o Statistical Analysis: The statistical analysis is robust, with appropriate use of the Mantel test and UPGMA clustering.
Discussion
• Interpretation of Results: The discussion interprets the results in the context of the research question and existing literature.
o Genetic Diversity: The study's findings on genetic diversity are well-discussed, highlighting the significance of the results for breeding programs.
o Comparison with Literature: The results are compared with previous studies, providing a comprehensive understanding of the findings.
o Limitations: The limitations of the study are acknowledged, and suggestions for future research are provided.
o Future Research Directions: The discussion outlines potential future research directions, emphasizing the importance of molecular techniques for further studies.
Conclusion
Summary of Findings: The conclusion succinctly summarizes the main findings of the study.
Implications for Breeding: The implications of the findings for commercial breeding programs and genetic diversity enhancement are discussed.
Recommendations: Recommendations for future research are provided, including the use of molecular techniques to examine specific morphological traits.
Figures and Tables
Clarity and Relevance: The figures and tables are clear, well-labeled, and relevant to the study.
o Figure 1 (Dendrogram): Effectively illustrates the genetic relationships among the genotypes.
o Table 1 (Materials): Provides a comprehensive list of the genotypes and hybrids used in the study.
o Table 2 (Descriptors): Descriptors for morphological characterization are well-defined and standard.
o Table 3 (Morphological Characteristics): Provides a detailed distribution of morphological characteristics, supporting the analysis of genetic diversity.
Raw Data
• Accessibility and Documentation: The raw data is accessible and well-documented, supporting the conclusions drawn in the manuscript.
Data Integrity: There are no concerns about data integrity or transparency. The data provided are robust and statistically sound.

---

## Round 0.2 · accepted · Accept

Your corrections, in accordance with the reviewer and editorial recommendations, are sufficient for the manuscript to be accepted. Congratulations